# Development of a breast cancer risk assessment and primary prevention pathway for women aged 30–39 years: Views of UK primary care providers on the role of primary care

**Sarah Hindmarch**[1], **Louise Gorman**[2], **Juliet A. Usher-Smith**[3], **Victoria G. Woof**[1], **Sacha J. Howell**[4], **David P. French**[1] *

1 Manchester Centre for Health Psychology, Division of Psychology and Mental Health, School of Health Sciences, Faculty of Biology, Medicine and Health, University of Manchester, Manchester, United Kingdom, 2 NIHR Greater Manchester Patient Safety Research Collaboration, Division of Population Health, Health Services Research & Primary Care, Faculty of Biology, Medicine and Health, University of Manchester, Manchester, United Kingdom, 3 Primary Care Unit, Department of Public Health and Primary Care, University of Cambridge, Cambridge, United Kingdom, 4 Division of Cancer Sciences, Faculty of Biology, Medicine and Health, Manchester Academic Health Science Centre, University of Manchester, Manchester, United Kingdom

* david.french@manchester.ac.uk

## Abstract

### Background

Identifying women aged 30–39 years at increased risk of developing breast cancer would allow them to receive screening and prevention offers. For this to be feasible, the practicalities of organising risk assessment and primary prevention must be acceptable to the healthcare professionals who would be responsible for delivery. It has been proposed that primary care providers are best placed to deliver a breast cancer risk assessment and primary prevention pathway. The present study aimed to investigate a range of primary care provider's views on the development and implementation of a breast cancer risk assessment and primary prevention pathway within primary care for women aged 30–39 years.

### Methods

Twenty-five primary care providers working at general practices in either Greater Manchester or Cambridgeshire and Peterborough participated in five focus groups (n = 18) and seven individual interviews. Data were analysed thematically and organised using a framework approach.

### Results

Three themes were developed. *Challenges with delivering a breast cancer risk assessment and primary prevention pathway within primary care* highlights that primary care are willing to facilitate but not lead delivery of such a pathway given the challenges with existing

**Data Availability Statement:** The data that support the findings of this study are openly available in Figshare at http://doi.org/10.48420/24058629.

**Funding:** Whilst conducting this work, SH was funded by a Manchester Cancer Research Centre PhD studentship. VGW is funded by a Medical Research Council PhD studentship (MR/N013751/1). DPF and SJH are supported by the NIHR Manchester Biomedical Research Centre (IS-BRC-1215-20007 and NIHR203308). JAU-S is funded by an Advanced Fellowship from the National Institute for Health and Care Research (NIHR300861). The views expressed are those of the authors and not necessarily those of the NHS, the NIHR or the Department of Health and Social Care. The funders had no role in study design, data collection and analysis, decision to publish, or preparation of the manuscript.

**Competing interests:** The authors have declared that no competing interests exist.

workload pressures and concerns about ensuring effective clinical governance. *Primary care's preferred level of involvement* describes the aspects of the pathway participants thought primary care could be involved in, namely co-ordinating data collection for risk assessment and calculating and communicating risk. *Requirements for primary care involvement* captures the need to provide a training and education package to address deficits in knowledge prior to involvement. Additionally, the reservations primary care have about being involved in the management of women identified as being at increased risk are discussed and suggestions are provided for facilitating primary care to take on this role.

## Conclusions

Despite optimism that primary care might lead a breast cancer risk assessment and primary prevention pathway, participants had a range of concerns that should be considered when developing such a pathway.

## Introduction

Pre-menopausal breast cancer incidence is increasing worldwide [1]. Younger women are more likely to develop aggressive breast cancer subtypes, which are associated with higher mortality despite intensive treatment regimens [2, 3]. As a result, breast cancer is the leading cause of death in women aged 35–50 years in the UK with 2,000 deaths reported per year [4]. There is, therefore, a need to identify younger women at increased risk of developing breast cancer who could benefit from earlier screening and preventive strategies [5].

In the UK, a strong family history of breast cancer or known high risk genetic variant in a close relative are the only criteria by which women aged under 50 years can access breast cancer screening and preventive strategies [6]. This approach relies on women either presenting with concerns about family history in primary care or having a family history identified during investigation of a breast problem in secondary care. However, at least 65% of women who develop breast cancer before the age of 50 years do not have a strong family history and are not currently identified as being at increased risk [7, 8]. The development of risk prediction models, such as the Tyrer–Cuzick and BOADICEA models, has made it possible to estimate an individual woman's risk of developing breast cancer with reasonable levels of accuracy [9, 10]. Each model differs on the 'classic' risk factors they incorporate into their risk estimates; this includes hormonal and reproductive risk factors, previous biopsies, and hereditary risk components (family history). More recently, risk prediction models have incorporated polygenic risk scores and mammographic breast density, which have improved their discriminatory accuracy [11, 12]. These models allow the tailoring of screening practices based on individual variation in risk (risk-stratified screening) and the opportunity for women to be offered preventive strategies such as weight loss or weight gain prevention interventions and risk-reducing medication.

It has been proposed by a number of researchers that primary care involvement in the delivery of a population wide risk-based screening and prevention programme is critical to its success, with their suggested role being to conduct the risk assessment and provide primary prevention advice in the form of discussing health behaviour change and risk-reducing medication [13–15]. A recent systematic review showed that primary care providers typically take a reactive role in breast cancer risk assessment that is predominantly focused on the collection of family history, and report infrequent use of multi-factorial risk assessment tools [16].

However, since June 2022, the proactive identification of women at increased risk of breast cancer within primary care has been facilitated by a change to National Institute for Health and Care Excellence (NICE) guidance whereby the recommendation to identify women only when they present with concerns was withdrawn [17]. Further, a recent review determined that proactive breast cancer risk assessment for women under 50 years within primary care currently satisfies many of the standard principles for screening but identified implementation into clinical practice as a key area of uncertainty [18].

The roles of individuals within a risk-based programme should be clearly defined and developed in conjunction with healthcare professionals themselves to ensure acceptability [19, 20]. Qualitative studies conducted in the UK have identified barriers and facilitators with respect to primary care involvement. However, these studies have focused on specific stages of the pathway required to provide breast cancer risk assessment and primary prevention advice, for example, the implementation of risk assessment [21–23], or risk-reducing medication [24]. Furthermore, only the views of general practitioners (GPs) and nurses have been captured in qualitative research to date. The primary care workforce in the UK has grown and diversified in recent years and now includes a wider range of allied healthcare professionals and health-care associate professionals such as pharmacists and physician associates who contribute to risk assessment and primary prevention activities. Therefore, it is important to examine the views of a range of primary care professions to understand how the wider workforce could support implementation considering declining GP numbers [25].

The present study aimed to explore primary care providers' views on the development and implementation of a breast cancer risk assessment and primary prevention pathway within primary care for women aged 30–39 years. Offering breast cancer risk assessment to this age group is necessary to ensure that maximum patient benefit can be realised through uptake of screening and preventive strategies from the age of 40 years. Specific objectives were to:

1. Understand primary care providers' views on involvement in breast cancer risk assessment and primary prevention activities

2. Identify perceived barriers and facilitators to implementation of a breast cancer risk assessment and primary prevention pathway within primary care

3. Identify the aspects of a prototype pathway that primary care providers consider more or less acceptable to be involved in

## Methods

### Design

A cross-sectional qualitative design primarily using focus groups was employed. Focus groups were considered the most appropriate method of data collection because they encourage lively debate and exploration of contradictions among members resulting in rich data [26]. They enable perspectives to evolve and become co-created, allowing insight into the degree of group consensus on the topic which was important given the objective of identifying which aspects of the prototype pathway are more or less acceptable [27]. One-to-one interviews were conducted when participants were unable to attend scheduled focus groups.

### Participants and setting

Participants were eligible if they were: (1) employed at a general practice within Greater Manchester or Cambridgeshire and Peterborough Integrated Care Systems, (2) a primary care

provider in one of the following professions: nurse, doctor, pharmacist, physician associate, nursing associate, or healthcare assistant, (3) able to provide informed consent, and (4) able to speak and understand English. We deliberately sought heterogeneity in profession amongst focus group participants to reflect current multidisciplinary professional practice. Group diversity encourages people to explain their reasoning revealing how the groupings construct their positions on the topic being examined [28]. This provides the opportunity to identify where the disputes and boundaries lie between different professions.

The locations were chosen so that pre-existing links the research team had with primary care contacts could be utilised given personal connections have been found to be a particularly effective way of recruiting primary care providers [29]. The patient populations in both geographical areas are different, with Greater Manchester having higher levels of deprivation and ethnic diversity than Cambridgeshire and Peterborough [30, 31].

## Procedure

Ethical approval was received from the University of Manchester Research Ethics Committee (Ref: 2019-7900-12761) and HRA and the study was carried out following the Good Clinical Practice principles and relevant regulations. All participants provided informed verbal consent to take part in the study prior to any study procedures. Verbal consent was obtained using a consent script and this was audio-recorded separately to the focus group or interview. This method of obtaining consent was approved by the University of Manchester Research Ethics Committee.

A variety of recruitment methods were used to target the eligible professions. Emails promoting the study were sent by the lead author to key individuals including clinical academic GPs known to the research team, a programme assistant at the North West GP training school, and primary care contacts at the Greater Manchester and Cambridgeshire and Peterborough Integrated Care Boards. Those contacted were asked to cascade a letter of invitation and participant information sheet across their primary care networks. Study advertisements were also circulated in primary care newsletters and shared on Twitter. Finally, snowball sampling was employed whereby participants shared the study information within their own professional networks to assist in identifying future participants. For all strategies, prospective participants were asked to contact the research team to express their interest in taking part and following confirmation of eligibility they were sent a participant information sheet if they had not already received one.

An initial draft of the topic guide was developed by the lead author, with questions guided by the aims of the study. Feedback on this draft was obtained from members of the research team (JAU-S, DPF and SJH) who have a wealth of clinical and research expertise in breast cancer and screening services, primary care and health services research, health psychology, and qualitative methods. The topic guide was pilot tested with a clinical academic GP who was not part of the research team, who provided feedback on wording and relevance of questions, prompts and flow. Questions focused on eliciting views about primary care involvement in breast cancer risk assessment and primary prevention, including perceived confidence in performing tasks, consideration of which primary care professions could be involved and perceived barriers and facilitators to involvement at each stage of the pathway (see S1 File). As the material to be discussed was likely to be unfamiliar to participants, background text was circulated as pre-reading material prior to each focus group and interview (see S2 File). During the focus groups and interviews, a prototype breast cancer risk assessment and primary prevention pathway was presented to participants to stimulate discussion about the potential role of primary care. Three components of breast cancer risk assessment were proposed in line with an

approach currently being offered to women aged 30–39 years in a Cancer Research UK funded feasibility study [32]: (1) a questionnaire for self-reported assessment of breast cancer risk factors, (2) a saliva sample for assessment of polygenic risk and mutations in high-risk genes, and (3) a low-dose mammogram for assessment of breast density. The topic guide was followed flexibly, allowing participants to raise issues important to their professional experience.

Online focus groups and interviews were conducted between 6th July 2022 and 10th March 2023, using Zoom video conferencing. Following consent, participants gave demographic information (gender, age, ethnicity, employment details (job title and years in current role) and practice postcode). For focus groups, participants provided this information to one of the facilitators, either via telephone or in a breakout room. Practice postcode was collected to assess the deprivation decile of the practice location using the Index of Multiple Deprivation 2019, a measure of relative deprivation for small areas in England [30]. Once the decile was extracted, the postcode was destroyed preserving the confidentiality of the participants. Data were audio-recorded and transcribed verbatim by an external transcription company. Identifiable information was removed from the transcripts, and participants were assigned a study ID consisting of their profession and a number. Data collection was primarily conducted by the lead author (SH) with another member of the research team taking notes and providing a verbal summary of the discussion prior to each focus group ending (VGW). Both facilitators were female doctoral students with a psychology disciplinary background, and at least five years postgraduate training and experience conducting qualitative research with healthcare professionals. Interviews were conducted by the lead author (SH). Data collection continued until the research team deemed that sufficient data had been collected to answer the research question [33]. A £50 cash payment and additional Internet expenses (£5 an hour) was offered to participants in recognition of their time given to the study.

## Data analysis

Data were analysed thematically and organised using a framework approach within NVivo 12 [34]. A critical realist approach was taken meaning we treated the data as indicating the participants' perception of their reality, which is shaped by and embedded within their cultural context, language and experiences [35, 36]. Primary data analysis was conducted by the lead author with input from LG, JAU-S, and DPF. Initial coding was deductive to ensure the study objectives were addressed. Additional codes were also developed inductively during data analysis to capture nuances in the data. The majority of coding was semantic, capturing explicitly expressed meaning and staying close to the language of participants. However, there was a shift towards more latent coding as the analysis progressed. The lead author constructed a working thematic framework of codes contained within categories which was reviewed and refined as coding progressed by way of regular meetings with LG and JUS before being applied to all remaining transcripts. The 'framework' feature in NVivo12 was used to plot each category onto a separate thematic matrix with codes presented in separate columns, and participants (cases) on separate rows. Data in each cell were summarized to synthesise the data set and illustrative extracts were noted. The resulting data in the framework matrices were compared across and within cases to highlight similarities and differences and develop initial themes. Regular meetings were held between members of the research team (SH, LG, JAU-S, and DPF) throughout analysis and interpretation to discuss explanations for findings, review and evaluate initial themes, and ultimately determine the overall thematic structure. The final thematic structure was agreed by the entire research team as representing participants' views.

## Results

A total of 25 primary care providers were recruited, with 18 taking part in focus groups (n = 2–5 per group) and seven taking part in interviews. Focus group duration ranged from 62 to 85 minutes (median 65 minutes) and interviews lasted between 48 and 65 minutes (median 54 minutes). Demographic and professional characteristics of the sample are presented in Table 1. The majority of participants were female (84%) and medically qualified (76%). Based on practice postcode, most participants worked in practices located in areas of high deprivation.

Data are presented as three themes with six sub-themes (see Table 2). Quotes are identified by participant profession type and number followed by interview (I) or focus group (FG) number.

### Theme 1: Challenges with delivering a breast cancer risk assessment and primary prevention pathway within primary care

**Sub-theme 1.1: Resolving existing workload and workforce capacity limitations.** Participants considered involvement in breast cancer risk assessment and primary prevention to be within the scope of primary care responsibilities. This was largely because involvement in risk assessment and primary prevention activities for other conditions, such as cardiovascular risk, was part of their routine practice. Despite this, given the challenges with existing workload pressures in the aftermath of the COVID-19 pandemic, all participants thought that primary care could facilitate but not lead the delivery of a breast cancer risk assessment and primary prevention pathway.

> *it's what are you going to give up, you know, so that you know, that's the decision. Yeah, of course we can do it, but are we not going to do flu vaccines or childhood immunisations, or cervical screening to make time for this as we can't do everything.*

> *(GP, 18, FG5)*

Participants were particularly concerned about the proportion of women who would be identified as at increased risk as management of these women would be more time consuming. To cope with this additional workload, some participants discussed the possibility of nominating a member of the practice team as a champion who would have protected time to manage women at increased risk should the workload implications warrant it.

By contrast, other participants discussed the option of delegating the additional workload across different professions within the primary care networks including practice-based roles such as physician associates and pharmacists. For example, pharmacists could be responsible for risk-reducing medication. Many participants viewed this positively and believed it would facilitate primary care involvement. However, one participant was concerned about variation in practitioner composition of the workforce across different general practices as this could lead to a lack of standardisation in delivery exacerbating existing health inequalities:

> *whilst it is very good to have this flexibility and we can devolve it to a lot of people working in the practice, for those small practices that don't have that luxury, I think—I mean, potentially, are they going to be in the more deprived areas anyway with less resources, and then there's isolated women in an isolated practice.*

> *(GP trainee, 5, FG2)*

**Table 1. Demographic and professional characteristics of participants (n = 25).**

| Characteristic | N (%) |
|---|---|
| **Gender** | |
| Female | 21 (84) |
| Male | 4 (16) |
| **Age (years)** | |
| 18–30 | 5 (20) |
| 31–40 | 12 (48) |
| 41–50 | 6 (24) |
| 51–60 | 2 (8) |
| **Ethnicity[a]** | |
| White British | 15 (60) |
| White (Other) | 2 (8) |
| White Irish | 1 (4) |
| Indian | 1 (4) |
| Pakistani | 1 (4) |
| Chinese | 1 (4) |
| Black (Other) | 1 (4) |
| Mixed (White & Asian) | 1 (4) |
| Other | 1 (4) |
| Prefer not to say | 1 (4) |
| **Profession** | |
| GP trainee | 10 (40) |
| GP | 9 (36) |
| Trainee nursing associate | 2 (8) |
| Healthcare assistant | 1 (4) |
| Physician associate | 1 (4) |
| Advanced nurse practitioner | 1 (4) |
| Advanced clinical pharmacist | 1 (4) |
| **Years in current role** | |
| 0–4 | 18 (72) |
| 5–9 | 3 (12) |
| 10–14 | 2 (8) |
| 15–19 | 1 (4) |
| ≥20 | 1 (4) |
| **Practice location** | |
| Greater Manchester | 23 (92) |
| Cambridgeshire or Peterborough | 2 (8) |
| **Deprivation decile of practice location[b]** | |
| 1–2 (most deprived) | 13 (52) |
| 3–4 | 6 (24) |
| 5–6 | 3 (12) |
| 7–8 | 3 (12) |
| 9–10 (least deprived) | 0 |

[a]Detailed ethnicity classifications reported in line with the current ethnicity harmonised standard [37]

[b]Derived from practice postcode using the Index of Multiple Deprivation 2019, a measure of relative deprivation for small areas in England [30]

**Table 2. Thematic structure.**

| Theme | Sub-theme |
|---|---|
| 1. Challenges with delivering a breast cancer risk assessment and primary prevention pathway within primary care | 1.1 Resolving existing workload and workforce capacity limitations |
| | 1.2 Ensuring effective clinical governance |
| 2. Primary care's preferred level of involvement | 2.1 Co-ordinating data collection for risk assessment |
| | 2.2 Calculating and communicating risk |
| 3. Requirements for primary care involvement | 3.1 Upskilling primary care providers |
| | 3.2 Overcoming reservations towards managing women at increased risk |

Ultimately, participants did not think delivery of a breast cancer risk assessment and primary prevention pathway in primary care was viable unless resourced appropriately in line with the expected increase in workload. The provision of financial incentives to calculate breast cancer risk such as inclusion in the quality and outcomes framework was perceived as necessary for increasing the likelihood of primary care performing the behaviour proactively.

*You'd need to offer incentives, I would have thought, especially when you think about the time it will take from the workforce*

*(Trainee nursing associate, 11, FG3)*

**Sub-theme 1.2: Ensuring effective clinical governance.** Participants expressed concerns about safely co-ordinating the delivery of a breast cancer risk assessment and primary prevention pathway within primary care. Their key concern focused on ensuring effective clinical governance arrangements with many participants preferring a centralised service delivery model akin to cancer screening programmes for this reason. Benefits of this delivery model were perceived to be robust quality assurance standards, a dedicated team to provide support to women and a systematic approach of following up on missing data to ensure accurate risk estimates. The latter was considered particularly important given the multifactorial nature of the risk assessment.

*I think I echo what other people have said, that it's [co-ordinating breast cancer risk assessment] a big risk to hold within primary care, where there's specialist services available*

*(Physician associate, 9, FG3)*

*if there was like three different chunks to it, um, you know, what happens if they don't bring back the saliva or what happens if they don't come in for their history bit, because then who would do that chasing? [. . .] how would that work in terms of getting all of it complete together so that you can get a meaningful result.*

*(GP, 24, I6)*

Furthermore, the multifactorial nature of the risk assessment was perceived as a logistical challenge to primary care co-ordinating the process of risk assessment. Participants questioned how results from the mammography component completed in secondary care could be accessed in primary care to allow risk to be calculated. On the contrary, one participant thought that not co-ordinating the process of risk assessment in primary care would be a

missed opportunity as the multifactorial nature of the risk assessment could be leveraged to identify additional information to breast cancer risk which would be useful for informing the patient's care:

> *I think the difficulty we'd have with not doing it in primary care is, even if we didn't deal with the women who were high risk of breast cancer, obviously, the measurements that you're taking can flag up a number of other issues. So, for instance, if they came back with a low risk of breast cancer but they had a really high cholesterol, or they turned out to be diabetic, then that will inevitably get bounced back to us anyway. (GP, 13, FG4)*A decentralised service delivery model co-ordinated through primary and secondary care was also discussed but largely dismissed. Participants were concerned that splitting the pathway across primary and secondary care could result in care fragmentation compromising the quality of care women receive.

### Theme 2: Primary care's preferred level of involvement

Despite concerns about leading the delivery of a breast cancer risk assessment and primary prevention pathway, participants highlighted which aspects of the pathway they felt primary care could most appropriately be involved in and provided suggestions for facilitating implementation.

**Sub-theme 2.1: Co-ordinating data collection for risk assessment.**   Much of the risk factor information required for a breast cancer risk assessment would already be captured in electronic medical records and participants felt the remaining information such as age at menarche could be collected easily by sending a questionnaire to patients' mobile phones. Participants described how this mode of communication has become commonplace following the COVID-19 pandemic. This was viewed by most participants as a better use of resources in comparison to taking up consultation time.

> *There is, and this is partly thanks to COVID, there is now much better ways of communicating with patients, like doing questions that you can just send them on their phones [. . .] They're getting cannier about responding to things online, we've been sending them links that we perhaps—you know, in the old days we might have given them a printout about something*
>
> *(GP, 19, I1)*

However, one participant was cautious of relying on a digital approach only to collect this information given her experience of speaking to many young women who did not own a mobile phone. Therefore, an alternative method of providing risk factor information would need to be considered such as an appointment with a healthcare assistant to ensure women from socioeconomically deprived areas have equitable access to risk assessment.

When considering potential opportunities for integrating components of the risk assessment into existing routine appointments, participants acknowledged the difficulty of doing so for the target population given limited points of contact; *it's actually a bit of an awkward demographic, isn't it, because they're too young for all of the chronic disease stuff, but they're too old for things like university checks (GP trainee, 15, FG5)*. Participants did not think it was appropriate to add a breast cancer risk assessment to cervical screening appointments due to this already being an anxiety provoking appointment for many women.

**Sub-theme 2.2: Calculating and communicating risk.**   Participants were similarly happy to calculate breast cancer risk providing a risk assessment tool similar to QRISK (cardiovascular disease risk assessment tool) was available and integrated into their IT system. To make the

process of calculating risk more efficient, it was considered particularly useful if the tool could automatically extract the relevant information coded in patient's medical records. Overall, participants would be willing to communicate the result of a breast cancer risk assessment if education for providers was available to help them interpret the risk result. Risk was recognised as an abstract and often difficult to understand concept for patients. Therefore, participants desired visual representations of risk such as icon arrays and a feature to demonstrate the impact of reducing risk on patient outcomes on the risk assessment tool to aid communication of the result:

*in terms of relaying the information to the patient, it's very easy to translate QRISK into case plots and use that visual, you know, representation of risk reduction, for maybe people who struggle with numbers.*

*(GP trainee, 4, FG2)*

### Theme 3: Requirements for primary care involvement

**Sub-theme 3.1: Upskilling primary care providers.** Whichever pathway was used to introduce breast cancer risk assessment, there was an expectation from participants that primary care providers would need to be upskilled to be involved in this new area of patient care. Participants were unfamiliar with the evidence base underpinning and supporting the introduction of breast cancer risk assessment for young women. Possibly due to this unfamiliarity, some participants expressed reservations about the robustness of the risk algorithm to accurately identify those at increased risk, particularly regarding accuracy of self-reported information and the stability of risk over time. Moreover, participants reported that availability of evidence demonstrating tangible differences in outcomes for young women participating in breast cancer risk assessment, such as numbers of lives saved, would make primary care more accepting of greater involvement:

*One of the things as well is I think that a lot of it relies on is how effective the whole thing is, so if you told me this will save of a population of women in my practice, this will save this many lives or you know, you know, you could convince me to get a lot more involved in it.*

*(GP trainee, 17, FG5)*

Participants did not feel they possessed sufficient knowledge about topics required to deliver a breast cancer risk assessment and primary prevention pathway; notably genetics, interpretation of risk results, and counselling about risk-reducing medication. Therefore, provision of a training and education package to equip primary care providers with the knowledge needed to answer patient queries about the pathway was perceived as essential prior to involvement.

*I can just think of the spectrum of questions that women are going to be asking about their risk, about different contraceptive choices, IVF, you know. [. . .] I can just imagine, a lot of the questions, we just might not have the answers to.*

*(GP trainee, 6, FG2)*

**Sub-theme 3.2: Overcoming reservations towards managing women at increased risk.** Most participants were comfortable with having health behaviour change conversations, with

many describing it as primary care's "bread and butter". However, participants in one focus group acknowledged the complexity of changing people's health behaviours and felt it would be more effective to signpost directly or via social prescribers to community health interventions for continued and more in-depth support than a primary care provider has the time or expertise to provide.

*because it is complex, you can't just give a one-off intervention to somebody and expect them to change so, you know, I would have thought that you probably need community services for those who are interested in changing to help coach them through it.*

*(GP, 18, FG5)*

The majority of participants who were prescribers expressed discomfort in discussing and prescribing Tamoxifen as a preventive medication; *I wouldn't touch it [Tamoxifen] with a bargepole now (GP trainee, 17, FG5)*. When prompted to consider the similarity with prescribing a statin to reduce cardiovascular risk, participants appreciated how the situations were analogous but expressed a reluctance to discuss and prescribe Tamoxifen. Reasons given for this reluctance included Tamoxifen prescribing not being part of routine practice in primary care and a lack of knowledge about its side effects and monitoring requirements to counsel women and manage Tamoxifen appropriately. Additionally, one participant believed that statins were underpinned by a more robust evidence base than Tamoxifen.

*I'm not familiar with things like side effects, monitoring [. . .] How do I know when to stop it, like which of the patients in the high-risk group would just be the lifestyle modification, which ones would be for Tamoxifen, those sorts of things like I don't know about. So, how can I prescribe safely if I don't know about it?*

*(GP trainee, 15, FG4)*

A major area of concern for many participants was the perceived appropriateness of prescribing Tamoxifen to the target population of women aged 30–39 years given it was believed to increase the risk of thrombosis and endometrial cancer.

*You are giving a drug to reduce the risk of something [. . .] but what is the actual absolute risk of increasing the endometrial cancer? Is it just a pointless exercise in some regards?*

*(GP trainee, 4, FG2)*

As family planning is likely to be a consideration for many women in this age group, some participants questioned the likelihood of women agreeing to take Tamoxifen (as an anti-oestrogen) due to its interference with childbearing.

Despite their concerns, participants made suggestions about how primary care could be made to feel more comfortable assuming responsibility for managing Tamoxifen. For example, if a specialist in secondary care wrote the first prescription or a NICE guideline was developed focused on the initiation and ongoing management of Tamoxifen for breast cancer risk reduction in the target population. Therefore, participants acknowledged that a change in practice was possible:

*I guess there are things that maybe ten, fifteen years ago that we would never have foreseen— that we wouldn't have felt comfortable doing in primary care, that we now do. So, you know, things do change, and a lot of sometimes our beliefs are held because of what we're*

*traditionally used to doing. [. . .] If you look at the management of diabetes, you know, that's transformed in the last ten years and we've got used to doing it.*

*(GP, 13, FG4)*

## Discussion

### Summary of main findings

The primary care providers taking part in this study viewed their involvement in a breast cancer risk assessment and primary prevention pathway for women aged 30–39 years as logical given it complemented their existing responsibilities with respect to risk assessment and prevention for other diseases. However, they did not think it was feasible, at present, for the pathway to be led by primary care given workload pressures and concerns about ensuring effective clinical governance within primary care. Collecting the risk factor information and calculating and communicating the risk result were considered the most acceptable stages for primary care to be involved in and management of women at increased risk the least. Provision of a training and education package was considered essential to facilitate primary care involvement in this new area of patient care.

### Relevance to existing literature

Previous literature has consistently identified primary care as the most opportune setting to conduct breast cancer risk assessment and provide prevention advice as part of the implementation of a risk-based screening and prevention programme [13–15]. However, the present findings suggest that primary care providers are concerned about leading such a pathway and are particularly cautious about assuming responsibility for the management of women identified as at increased risk. In line with previous research [16, 24], this study found that primary care providers are particularly uncomfortable with the prospect of discussing and prescribing Tamoxifen as a breast cancer risk-reduction strategy. However, the present study also offers novel insight into perceived concerns about offering Tamoxifen to young women. Participants were concerned that Tamoxifen would increase the risks of thrombosis and endometrial cancer. Although the risk of thrombosis is increased, the absolute risk remains low in young women being comparable to treatment with the combined oral contraceptive. The increase in endometrial cancer risk is only observed when Tamoxifen is used by post-menopausal women [38]. Therefore, concerns about Tamoxifen use in this population could be alleviated through provision of education.

Although such education could be provided to facilitate primary care involvement in risk management, it may be challenging to justify the considerable resources needed to upskill primary care providers to take on risk management if few women would be eligible for this per practice. If this were the case, it may be more appropriate for primary care to identify women at increased risk and refer them to a Family History, Risk and Prevention Clinic for discussion of risk management strategies. This approach would also be in line with participants' preferred level of involvement.

A key consideration underpinning participants' views about involvement in breast cancer risk assessment and primary prevention was the expected impact on workload given existing workload pressures within primary care. The diversity of the primary care workforce was perceived as a facilitator to their involvement as additional workload could be delegated across different professions. However, regional differences in the composition of the primary care workforce in England have been found, with integration of allied healthcare and healthcare

associate professionals lacking [39]. Therefore, a service delivery model reliant on contributions from these professionals has the potential to introduce or exacerbate existing health inequalities. Nonetheless, practice-based pharmacists contribute towards detailed tasks related to medicine prescription, reducing the burden of these activities for existing staff and improving prescribing practices [40, 41]. This suggests that pharmacists could play an important role in facilitating primary care involvement in risk-reducing medication.

In the present study, the multifactorial nature of the risk assessment was also perceived as a barrier to primary care co-ordinating the process of risk assessment. Participants were concerned about ensuring effective clinical governance arrangements for breast cancer risk assessment because of the risk of missing data and implications of this for the accuracy of risk estimates. Additionally, how the results from the mammography component would be accessible by primary care to enable risk calculation was unclear to participants. Mammographic density is a well-established risk factor for breast cancer in older women but its contribution to risk in women aged 30–39 years remains unknown. A study is currently ongoing to determine the magnitude of breast cancer risk associated with mammographic density in women aged 30–39 years [32]. Depending on the results of this study, assessment of mammographic density may well be necessary so how this could be co-ordinated within primary care would be important to consider.

## Strengths and limitations

This study elicited primary care providers' views across a breast cancer risk assessment and primary prevention pathway. This is considered a strength of the study as a more comprehensive understanding of involvement was achieved compared to previous research which has been limited in scope to specific stages of the pathway. Furthermore, we recruited a range of professionals including allied healthcare and healthcare associate professionals, with varying levels of clinical experience. This facilitated a more holistic understanding of the issues affecting primary care involvement in breast cancer risk assessment and primary prevention as staff were able to offer their views from the perspective of their differing roles. However, we acknowledge that the views of the nursing profession were underrepresented, and many participants were relatively new to their posts.

Recruitment to focus groups proved challenging given the time commitment required to participate. An alternative approach could have been to recruit multiple members of staff from the same practice so that a team meeting could be utilised for the focus group. This approach may have resulted in more solution focused dialogue as they would be able to reflect on their own experiences as a working group to inform how the pathway could work within their own practice. Nevertheless, those who did partake in focus groups brought a range of perspectives and disagreements were voiced which may not have been the case if participants had known each other.

Participants were recruited from two regions with diversity in their patient populations in terms of socioeconomic status and ethnicity. However, they are both in England which limits the transferability of the findings as countries vary substantially in how primary healthcare is organised and delivered [13]. For example, in the USA, providers specialising in women's health are considered part of the primary care workforce. A recent review found that women's health providers report feeling more comfortable with respect to both breast cancer risk assessment and primary prevention, resulting in greater reported use of risk assessment tools [16]. Therefore, there may be fewer challenges with implementing a breast cancer risk assessment and primary prevention pathway in primary care in the USA or other countries with different primary care systems.

## Implications and future research directions

The present study has identified that primary care providers are willing in principle to be involved in the delivery of a breast cancer risk assessment and primary prevention pathway for young women, but a range of concerns were identified. Future research should focus on developing and evaluating strategies for implementing multifactorial risk assessment within primary care, taking into account the need to minimise any additional workload, potentially through the use of additional allied healthcare professionals, the strong preference for governance of any pathway to sit outside individual primary care practices, and provision of adequate training, particularly around the prescription of Tamoxifen. Options include, for example, integrating risk assessment with consultations led by pharmacists, allied healthcare professionals or nurses related to prescription/review of contraception and cervical cancer screening. However, concerns have been raised about causing additional anxiety to these already anxiety-provoking appointments [21]. These approaches should be piloted to ascertain whether these concerns are well-founded. These approaches should be piloted to ascertain whether concerns about causing additional anxiety to these already anxiety-provoking appointments are well-founded. Furthermore, better modelling and communication of the numbers of women involved per general practice is needed to inform the scope of primary care's involvement.

## Conclusion

Despite optimism that primary care might lead a breast cancer risk assessment and primary prevention pathway, participants had a range of concerns that should be considered when developing such a pathway.

## Supporting information

**S1 File. Focus group and interview topic guide.**
(DOCX)

**S2 File. Pre-reading material.**
(DOCX)

## Acknowledgments

We would like to thank all the participants who kindly gave their time to take part in this study. We would also like to thank Brian McMillan for reviewing the topic guide and advising on recruitment strategies.

## Author Contributions

**Conceptualization:** Sarah Hindmarch, Louise Gorman, Juliet A. Usher-Smith, Sacha J. Howell, David P. French.

**Data curation:** Sarah Hindmarch.

**Formal analysis:** Sarah Hindmarch, Louise Gorman, Juliet A. Usher-Smith, David P. French.

**Funding acquisition:** Sacha J. Howell, David P. French.

**Investigation:** Sarah Hindmarch, Victoria G. Woof.

**Methodology:** Sarah Hindmarch, Louise Gorman, Juliet A. Usher-Smith, Sacha J. Howell, David P. French.

**Project administration:** Sarah Hindmarch.

**Supervision:** Louise Gorman, Sacha J. Howell, David P. French.

**Visualization:** Sarah Hindmarch.

**Writing – original draft:** Sarah Hindmarch.

**Writing – review & editing:** Sarah Hindmarch, Louise Gorman, Juliet A. Usher-Smith, Victoria G. Woof, Sacha J. Howell, David P. French.

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
