## [Decision Letter · Decision Letter 0]

13 Jun 2024

PONE-D-24-10167Development of a breast cancer risk assessment and primary prevention pathway for women aged 30-39 years: views of UK primary care providers on the role of primary carePLOS ONE

Dear Dr. French,

Thank you for submitting your manuscript to PLOS ONE. After careful consideration, we feel that it has merit but does not fully meet PLOS ONE’s publication criteria as it currently stands. Therefore, we invite you to submit a revised version of the manuscript that addresses the points raised during the review process.

**The paper tackles an interesting topic that is useful to a global audience. The language and grammar still need editing and proof-reading. Also kindly explain the thematic analysis process as pointed out by the reviewer and explain the implications of your work to the global readership.**

We look forward to receiving your revised manuscript.

Kind regards,

Aloysius Gonzaga Mubuuke

Academic Editor

PLOS ONE

Journal Requirements:

Additional Editor Comments:

The paper is an interesting one. In addition the review comments, the authors need to explicitly state the implications of their work to a global audience within the discussion and also proof-read the paper entirely.

Reviewers' comments:

Reviewer's Responses to Questions

**Comments to the Author**

1. Is the manuscript technically sound, and do the data support the conclusions?

Reviewer #1: Yes

Reviewer #2: Yes

2. Has the statistical analysis been performed appropriately and rigorously? 

Reviewer #1: N/A

Reviewer #2: Yes

3. Have the authors made all data underlying the findings in their manuscript fully available?

Reviewer #1: Yes

Reviewer #2: Yes

4. Is the manuscript presented in an intelligible fashion and written in standard English?

Reviewer #1: Yes

Reviewer #2: Yes

5. Review Comments to the Author

**Reviewer #1: **Thank you for inviting me to review this paper. The paper presents important qualitative data on the views of primary care providers with respect to pro-active identification of women at higher risk of cancer under the current screening age. The paper is well written, with a clear rationale, well justified methods and appropriate results and conclusions. The results provide useful and comprehensive views from healthcare professionals that will be very informative in the field as risk assessment is implemented. The limitation section also provides interesting reflection on the strengths and limitations of the study design that will be useful for future researchers.

One minor suggestion for improvement below:

Line 74 mentions a change in NICE guidelines – could you briefly mention exactly what the change entailed?

**Reviewer #2: **Strengths

1. Relevance and Importance: The topic is highly relevant, considering the increasing incidence of pre-menopausal breast cancer and the potential benefits of early risk assessment and prevention.

2. Methodology: The use of focus groups and individual interviews allows for a comprehensive understanding of primary care providers' views.

3. Clear Presentation of Findings: The manuscript is well-structured and well-written.

Major Concerns

1. Generalizability: The study is limited to primary care providers in Greater Manchester and Cambridgeshire and Peterborough. It would be beneficial to discuss the potential for generalizing these findings to other regions or countries.

2. Detailed Methodology: The manuscript would benefit from a more detailed description of the thematic analysis process, including coding procedures and how themes were identified and validated.

3. Implementation Feasibility: While the manuscript discusses the challenges and preferred involvement of primary care providers, it could further elaborate on specific strategies or frameworks for integrating this pathway into existing primary care practices.

Minor Concerns

1. Literature Review: The introduction could provide a more comprehensive review of existing risk prediction models and their current use in primary care.

2. Clarity in Results: Some of the quotes from participants could be more clearly linked to the themes and subthemes to enhance the coherence of the results section.

Recommendations

1. Improve on Generalizability: Add detailed information on how the findings might apply to primary care settings outside the study locations.

2. A clearer description of the Methodology: The thematic analysis process should be described in more detail.

3. Detail Implementation Strategies: Suggest concrete frameworks or strategies for integrating the risk assessment and prevention pathway into primary care.

4. Link Quotes to Themes: Link participant quotes to the relevant themes and subthemes.

5. Incude more details under Ethical Considerations: Write clearer on how participant confidentiality was ensured.

6. PLOS authors have the option to publish the peer review history of their article (what does this mean?). If published, this will include your full peer review and any attached files.

Reviewer #1: No

Reviewer #2: **Yes: **Oluchi Kanma-Okafor

---

## [Author Response · Author response to Decision Letter 0]

15 Jul 2024

Please see full response to all reviewer comments in the document attached

---

## [Editor Report · Decision Letter 1]

29 Jul 2024

Development of a breast cancer risk assessment and primary prevention pathway for women aged 30-39 years: views of UK primary care providers on the role of primary care

PONE-D-24-10167R1

Dear Dr. French,

We’re pleased to inform you that your manuscript has been judged scientifically suitable for publication and will be formally accepted for publication once it meets all outstanding technical requirements.

Kind regards,

Aloysius Gonzaga Mubuuke

Academic Editor

PLOS ONE

Additional Editor Comments (optional):

None
---

## [Editor Report · Acceptance letter]

2 Aug 2024

PONE-D-24-10167R1 

PLOS ONE

Dear Dr. French, 

I'm pleased to inform you that your manuscript has been deemed suitable for publication in PLOS ONE. Congratulations! Your manuscript is now being handed over to our production team.

Kind regards, 

on behalf of

Dr. Aloysius Gonzaga Mubuuke 

Academic Editor

PLOS ONE